# Does Myasthenia Gravis Affect Long-Term Survival in Thymic Carcinomas? An ESTS Database Analysis

**DOI:** 10.3390/diagnostics12071764

**Published:** 2022-07-21

**Authors:** Filippo Lococo, Dania Nachira, Marco Chiappetta, Jessica Evangelista, Pierre Emmanuel Falcoz, Enrico Ruffini, Paul Van Schil, Marco Scarci, Jòzsef Furàk, Francesco Sollitto, Francesco Guerrera, Lorenzo Spaggiari, Clemens Aigner, Liverakou Evangelia, Andrea Billè, Bernhard Moser, Pascal Alexandre Thomas, Moishe Liberman, Souheil Boubia, Alessio Campisi, Luca Ampollini, Alper Toker, Attila Enyed, Luca Voltolini, Dirk Van Raemdonck, Stefano Margaritora

**Affiliations:** 1Thoracic Surgey, Università Cattolica del Sacro Cuore, 00168 Rome, Italy; filippo.lococo@policlinicogemelli.it (F.L.); dania.nachira@policlinicogemelli.it (D.N.); evangelistajessica664@gmail.com (J.E.); stefano.margaritora@policlinicogemelli.it (S.M.); 2Thoracic Surgery, Fondazione Policlinico Universitario A. Gemelli IRCCS, 00168 Rome, Italy; 3Department of Surgical Sciences, Strasbourg University Hospital, 67000 Strasbourg, France; pefalcoz@gmail.com; 4Department of Surgical Sciences, University of Torino, 10124 Torino, Italy; enrico.ruffini@unito.it (E.R.); francesco.guerrera@unito.it (F.G.); 5Department of Thoracic and Vascular Surgery, Antwerp University Hospital, Antwerp University, 2000 Antwerp, Belgium; paul.vanschil@uza.be; 6Thoracic Surgery Divison of Cardiothoracic Surgery, Imperial College NHS Healthcare Trust, Du Cane Road, London W12 0HS, UK; marco.scarci@mac.com; 7Department of Surgery, University of Szeged, Semmelweis u. 8, H-6725 Szeged, Hungary; jfurak@gmail.com; 8Thoracic Surgery, Università di Foggia, 71122 Foggia, Italy; francesco.sollitto@unifg.it; 9Division of Thoracic Surgery, IEO, European Institute of Oncology, IRCCS, 20139 Milan, Italy; lorenzo.spaggiari@ieo.it; 10Department of Thoracic Surgery, Essen University Hospital—Ruhrlandklinik, 45239 Essen, Germany; clemens.aigner@rlk.uk-essen.de; 11Thoracic Surgery Department, Evangelismos Hospital, 106 76 Athens, Greece; kostas.antonopoulos@gmail.com; 12Thoracic Surgery Department, Guys Hospital London, London SE1 9RT, UK; andrea.bille@gstt.nhs.uk; 13Department of Thoracic Surgery, Medical University of Vienna, 1090 Vienna, Austria; bernhard.moser@meduniwien.ac.at; 14Department of Thoracic Surgery, North Hospital Aix-Marseille University, 13015 Marseille, France; pascalalexandre.thomas@ap-hm.fr; 15Division of Thoracic Surgery—Centre Hospital de l’Universite de Montreal, University of Montreal, 900 Rue Saint-Denis, Montreal, QC H2X 0A9, Canada; moishe.liberman@umontreal.ca; 16Department of Thoracic Surgery, Universitary Hospital Ibn Rochd Casablanca, Casablanca 20250, Morocco; souheilboubia@yahoo.fr; 17Cardiovascular and Thoracic Department, University of Verona School of Medicine and Verona University Hospital Trust, 37126 Verona, Italy; alessio.campisi@studio.unibo.it; 18Unit of Thoracic Surgery, Azienda Ospedaliera-Universitaria di Parma, 43126 Parma, Italy; luca.ampollini@unipr.it; 19Department of Cardiovascular and Thoracic Surgery, West Virginia University School of Medicine, Morgantown, WV 26506, USA; salpertoker@gmail.com; 20Department of Surgery, Faculty of Medicine, University of Debrecen, 22 Moricz Zsigmond Str., H-4032 Debrecen, Hungary; drenyediattila@gmail.com; 21Thoracic Surgery Unit, Careggi University Hospital, 50134 Florence, Italy; luca.voltolini@unifi.it; 22Department of Thoracic Surgery, University Hospitals Leuven, 3000 Leuven, Belgium; dirk.vanraemdonck@uzleuven.be

**Keywords:** thymic carcinoma, myasthenia gravis, surgery, recurrence

## Abstract

Background: Thymic carcinoma is a rare and highly malignant tumor with a dismal prognosis, which occasionally coexists with myasthenia gravis (MG). This study aims to investigate the MG incidence on a surgical cohort of patients with thymic carcinoma and to explore its influence on long-term survival. Methods: the prospectively collected data from the ESTS database on thymic epithelial tumors were reviewed. Clinical, pathological, and survival information on thymic carcinoma were analyzed. Results: the analysis was conducted on 203 patients, with an equal gender distribution (96 males and 107 females). MG was detected in 22 (10.8%) patients, more frequently elderly (>60 years, *p* = 0.048) and male (*p* = 0.003). Induction therapy was performed in 22 (10.8%) cases. After surgery, 120 (59.1%) patients had a Masaoka stage II–III while complete resection (R0) was achieved in 158 (77.8%). Adjuvant therapy was performed in 68 cases. Mean follow-up was 60 (SD = 14) months. The 3-year, 5-year and 10-year survival rates were 79%, 75% and 63%, respectively. MG did not seem to influence long-term survival (5-year survival in non-MG–TCs 78% vs. 50% in MG–TCs, *p* = ns) as age < 60 years, female gender, early Masaoka stage, and postoperative radiotherapy did, conversely. Conclusions: myasthenia occurred in about 10% of thymic carcinomas and it did not seem to affect significantly the long-term prognosis in surgically treated thymic carcinoma-patients.

## 1. Introduction

Primary thymic epithelial tumors are rare, but account for 20% of all mediastinal tumors [1]. While the overall incidence of thymic neoplasms has been estimated to be 0.13/100,000 (1% of all adult cancers), thymic carcinomas (TCs) are extremely rare and constitute only 0.06% of all malignancies [2], usually occurring in the fifth decade of life [3].

TCs can be distinguished from thymomas based on their differing histological, genetic, clinical, and radiological features. A robust body of evidence suggests that TCs have an aggressive biological behavior and generally present as a locally advanced/metastatic disease [4,5]. Consequently, TCs have a much poorer prognosis than thymomas with a 5-year survival estimated to be less than 50% for thymic carcinomas [5]. Surgery seems to play a crucial role in the treatment of TC-patients (generally in combination with chemotherapy and radiotherapy) but besides Masaoka stage and completeness of resection other prognostic factors have not clearly been validated.

Interestingly, while 30–40% of thymomas [6] are accompanied by myasthenia gravis (MG)―an autoimmune disease clinically characterized by fatigable weakness―the coexistence of MG and TC seems to be less frequent (<15%) [7,8]. Moreover, the prognostic role of MG is almost unclear in thymoma and even more in TCs.

Conflicting opinions on this topic have been reported. Some authors suggest a protective role of MG in TC-patients due to early diagnosis in patient presenting MG-related symptoms or to the assumption that TC with MG may have specific pathological features making the tumor less aggressive, and thus more responsive to treatment [9]. However, this finding was not confirmed in other series [8,9,10].

The aims of the present study were to investigate the overall incidence of MG in this surgical cohort of TC patients, then the association between MG and other clinico-pathological parameters and finally to explore (among others) its influence on long-term survival after surgery.

## 2. Materials and Methods

An observational, multicenter, retrospective study was conducted using data collected in the ESTS database on surgically treated thymic epithelial tumors. Among the records of 2506 patients operated on from 1/2000 to 7/2019, the data of patients with a pathological diagnosis of TCs were extracted.

Detailed clinical, surgical, pathological, and survival information of patients operated for TCs were reviewed. In particular, all data about MG and non-MG patients with thymic carcinoma were analyzed.

The present study was approved by the ESTS Database Committee and by the ethics commission of promoting center (2020/8903), and was therefore performed in accordance with the ethical standards of the Declaration of Helsinki and its later amendments. Individual informed consent was necessarily waived due to the retrospective nature of this study and to the anonymity of patients enrolled.

All thymectomies were performed under general anesthesia and double or single lung ventilation according to the surgical approach. The most adopted approaches in the centers involved in the ESTS database were sternotomy and thoracotomy, followed by robotic (RATS) and video-assisted thoracic surgery (VATS) accesses or clamshell/hemi-clamshell, based on local surgical expertise and clinical decision based on tumor stage.

Oncological follow-up of patients was conducted through medical examination (by oncologists and/or thoracic surgeons) and radiographic evaluation according to the internal policy of the centers involved.

### Statistical Analysis

All categorical variables were reported as absolute numbers and percentages (%), continuous variables as mean followed by standard deviation (SD) in brackets. Kolmogorov–Smirnov test was used to evaluate normal distribution of data. Categorical variables were compared by Chi-squared test; continuous variables by independent sample Student’s *t*-test if normal distributed or by Mann–Whitney U-test if not normal.

In order to reduce biases, the analysis was performed evaluating specific and almost complete items (age, gender, MG, surgical access, Masaoka stage, surgical radicality, neoadjuvant and adjuvant therapy, overall survival and mortality) from a minimum dataset (only the covariates ECOG, neoadjuvant/adjuvant therapy and surgical access presented missing values, Table 1).

Furthermore, to assess the effect of missing data on results, a sensitivity analysis of augmented data sets using Bayesian Multiple Imputation by Chained Equations (MICE) [11] was performed. The missing completely at random (MCAR) assumption was evaluated and confirmed with Little’s test; then a multiple imputation analysis (5 imputed datasets) was run with the expectation maximization method.

Overall survival (OS) was defined as time elapsed between surgery and death for cancer.

Survival function was estimated by Kaplan–Meier curves for the whole population and comparison of survivals stratified for different clinical categorical variables (sex, type of surgical approach, MG, age > 60 years, Ro resection, etc.) was performed by Log-rank test.

Univariable analysis was performed by Cox regression model. Any variable with a *p*-value less than 0.20 at univariable analysis was included in a Cox proportional hazards regression model to investigate the adjusted effect of an independent variable (MG, female gender, age < 60 years, early Masaoka stage, R0 resection, adjuvant radiotherapy) on long-term survival.

A *p*-value less than 0.05 was considered statistically significant. When a subgroup analysis was performed (patients younger than 60 years or patients undergone R0 thymectomy), alpha levels were adjusted by Bonferroni’s correction (simplified formula: apha/k, where k is the number of tests or pairs).

Statistical analysis was performed using IBM SPSS Statistics for Macintos, Version 25.00 (Armonk, NY, USA).

## 3. Results

There were 203 patients that underwent surgery for thymic carcinoma in the ESTS database. Ninety-six (47.3%) were males and 107 (52.7%) females. The mean age of the population was 59.7 (SD = 13.8; range 20–91) years. MG was detected in 22 (10.8%) TCs, more frequently in elderly (>60 years, *p* = 0.048) and male (*p* = 0.003) patients. The main clinical and pathological characteristics of the whole cohort and MG group are reported in Table 1. Although no data was available on the onset of MG in relation with TC occurrence, MG was never observed as post-surgical complication after thymectomy.

The mean major diameter of the lesion on preoperative CT-scan was 5.7 (SD = 3.5) cm. Induction therapy was performed in 22 (10.8%) cases. In 78 (38.4%) cases, the thymectomy was performed through a sternotomic access, followed by a thoracotomy in 33 (16.3%) cases. Minimally invasive approaches, like RATS or VATS were used in 17 (8.4%) and 10 (4.9%) cases, respectively.

The mean diameter of the lesion measured on specimen was 2.4 (SD = 0.7 cm). At pathological staging, most patients were classified as Masaoka stage II–III (124 patients, 61.1%), 53 (26.1%) as Masaoka stage I, and 26 as Masaoka stage IV.

A complete resection (R0) was achieved in 158 (77.8%) cases. Adjuvant therapy was performed in 68 cases (more specific: radiotherapy in 60 cases, chemotherapy in 5 and a combined radio-chemotherapy in 3).

The mean follow-up was 60.0 (SD = 14.0) months. The 3-year, 5-year, and 10-year survival rates were 79%, 75%, and 63%, respectively (Figure 1).

The type of surgical approach, minimally invasive (RATS or VATS) or traditional (thoracotomy or sternotomy) did not influence OS (*p* = 0.67). The concomitant presence of MG does not significantly impact on long-term survival in overall population (5-year survival in non-MG–TCs was 78.0% vs. 50.0% in MG–TCs; HR [95% CI]:1.9 [0.9–4.5], *p* = 0.096) and when evaluating only patients younger than 60 years (HR [95% CI]:2.7 [0.8–9.2], *p* = 0.083 (Bonferroni adjusted *p*-value = 0.025)).

On the contrary, age < 60 years (5-year survival: 82% vs. 70%, *p* = 0.039), female gender (5-year survival: 80% vs. 68%, *p* = 0.063), early Masaoka stage (5-year survival: 60% in Masaoka stage IIa vs. 24% in Masaoka stage IVb, *p* = 0.04, Figure 2A), R0-resection (5-year survival: 98% vs. 71%, *p* = 0.006, Figure 2B), and postoperative radiotherapy (5-year survival: 74% vs. 55% for CT, *p* = 0.003) were associated with better OS (Table 2).

At Cox regression analysis, age < 60 years (HR [95% CI]: 0.1 [0.0–0.4], *p* = 0.001), female gender (HR [95% CI]: 0.4 [0.2–1.0], *p* = 0.05), Masaoka stage ≤ IIa (HR [95% CI]: 0.4 [0.2–0.8], *p* < 0.001), and postoperative radiotherapy (HR [95% CI]: 0.5 [0.2–0.8], *p* < 0.001) were confirmed to positively affect long-term survival (Table 2), while MG not (HR [95% CI]: 8.7 [0.8–95.3], *p* = 0.07).

Analyzing only the subset of 158 patients who underwent a radical (R0) thymectomy for TCs, 72 (45.0%) were males, 92 (57.5%) > 60 years and 18 (11.3%) affected by MG; only 15 (9.4%) patients underwent a preoperative neoadjuvant therapy, 3 (1.9%) an adjuvant RT and 46 (28.7%) adjuvant CT.

The main variables influencing survival at univariable analysis were age > 60 years (5-year survival: 90% vs. 71%, *p* = 0.007) and advanced Masaoka stage (5-year survival: 60% in Masaoka stage IIa vs. 22% in Masaoka stage IVb, *p* = 0.04). Cox proportional hazards model confirmed the positive effect of age < 60 years (HR [95% CI]: 0.1 [0.0–0.5], *p* = 0.001) and Masaoka stage ≤ IIa (HR [95% CI]: 0.7 [0.2- 3.9], *p* = 0.001) on long-term survival, as reported in Table 3. Female gender (HR [95% CI]: 0.3 [0.1–0.8], *p* = 0.017) was at the threshold of the Bonferroni adjusted *p*-value (0.017) for this subgroup analysis 3.2.

## 4. Discussion

The present study based on the ESTS thymic database aims firstly to investigate the overall incidence of MG in this surgical cohort of TC patients, then the association between MG and other clinico-pathological parameters and finally to explore (among others) its effect on long-term survival after surgery.

### 4.1. Association between MG and TC and Clinical Features

While the association between MG and thymomas is clearly investigated, with an incidence ranging from 30–65% [5], controversial results have been reported on MG as a paraneoplastic syndrome of TC. In a recent review, Syrios et al. [3] stated that TCs are not associated with paraneoplastic phenomena, such as MG. On the contrary, Li et al. [9] assumed that it can still be considered as a paraneoplastic syndrome of TC, even if the incidence reported in the literature is quite low (<15%) [7,8,9,10].

In the present study, 22 TC patients (10.8%) were affected by MG, suggesting the rare but possible association between TCs and this neurological paraneoplastic syndrome. We observed that MG occurred more frequently in elderly (>60 years, *p* = 0.048) male (*p* = 0.003) patients. These data could be of interest considering that thymoma-associated MG patients were younger (52.0 years in [12]) and were equally reported in males and females [6].

We did not find any association with smaller tumor size, earlier Masaoka stage and a higher chance for radical resection, as reported by Li et al. (but their assumptions were based on a series of only 6 TC-patients with MG). Unfortunately, we have no data available in the ESTS database on the neurological results after surgical treatment in this population.

### 4.2. Long-Term Survival Results

Clinical and pathological characteristics of TC patients at surgery were in line with those reported in literature. In detail, the tumor-size, Masaoka stage (see Table 1), and percentage of patients who underwent adjuvant therapy (41%) confirmed the results already reported in previous reviews on this topic [5,6]. Regards of the radical resection rate (≈78% in the present analysis) this seems to be higher than that reported in other experiences (see Table 4), as a consequence of a selection bias of the ESTS database (see below).

Exploring long-term survival in TC-patients registered in the ESTS database, we observed 3-year, 5-year, and 10-year survival rates of 79%, 75%, and 63%, respectively. Substantially in line with our results, Fu et al. [13], analyzing long-term results of a large-sample multicenter database in China (“Chinese Alliance for Research of Thymoma”), reported 3-year, 5-year, and 10-year OS rates of 78.3%, 67.1%, and 47.9%, respectively. Worse results were observed analyzing 154 TCs registered in the Japanese Association for Chest Surgery register [14], (3-year, 5-year, and 10-year survival rates of ≈60%, ≈50%, and ≈40.0%, respectively) and other series summarized in Table 4. To explain the better long-term survival results reported in the present analysis, we should keep in mind that the ESTS database is a surgical collection of thymic epithelial tumor patients. Differently from other registries [13,14,15], the ESTS database collected only surgical cases coming from high-volume experienced institutions over the world (selection bias), this also reflected the high proportion of R0-resection (as reported above).

The present analysis showed that an age of less than 60 years (*p* = 0.039), early Masaoka stage (*p* = 0.005), R0-resection (*p* = 0.006), and postoperative radiotherapy (*p* = 0.003) were associated with better overall survival.

Concerning the influence of age on long-term survival, this might represent a selection bias (very frequent in all surgical series), considering that younger TC-patients are more likely to be treated with a combination of surgery and adjuvant therapy. The impact of Masaoka stage, surgical radicality, and adjuvant therapy on survival results have been largely reported in other series [8,13,14] summarized in Table 4.

In our analysis, early Masaoka stage (5-year survival: 60% in Masaoka stage IIa vs. 24% in Masaoka stage IVb, *p* = 0.04), R0-resection (5-year survival: 98% vs. 71%, *p* = 0.006, Figure 2B), had a strong positive impact on long-term survival, confirming the crucial role of surgery in the TC-patients care strategies. We also found that between patients who underwent adjuvant therapy, radiotherapy patients had better long-term results suggesting that, as in TCs, local disease control is a crucial element.

### 4.3. MG and Long-Term Survival

Recent evidence suggests that MG is not an independent or direct prognostic factor in thymoma patients [15,16] while few data are available for TC patients.

In the present analysis, the concomitant presence of MG does not influence long-term survival at univariable (5-year survival in non-MG–TCs 78% vs. 50% in MG–TCs, HR [95% CI]:1.9 [0.9–4.5], *p* = 0.096) and multivariable (HR [95% CI]: 8.7 [0.8–95.3], *p* = 0.07) analyses.

This result was in line with previous data [8,10,11]. However, Li et al. [9] suggested a protective role of MG in TC-patients. The authors explored in detail the prognostic role of MG in 49 TC-patients (6 affected by MG also) and observed, at univariable analysis, an association between the presence of MG and a better long-term survival. However, the number of patients was very low and this association was not confirmed at multivariable analysis. Moreover, the different theories postulated by the authors to explain this result are rather weak. Indeed, even if from a theoretically point of view a larger number of TC could benefit from early diagnosis (anticipated by the onset of MG-related symptoms), this assumption is not confirmed in the literature [8,9,10], including our work, in which no association between early Masaoka stage and MG was observed.

Similarly, the assumption that TC in patients with MG may have specific pathological features making it less aggressive and thus more responsive to treatment is far from being demonstrated. In this setting, only a focused study exploring the histological and bio-molecular features of MG–TC vs. non-MG–TC could clarify this point.

### 4.4. Limitations and Points of Strength

This study had several limitations and strengths. This was a retrospective multi-institutional study with clinical data missing in a small part of the sample. Moreover, there is a double selection bias. Firstly, only surgical cases have been included in the analysis; accordingly, the present population of study may not be representative of the entire population of TCs. Clinico-pathological variables affecting survival in this study should be interpreted with caution and should be considered only after surgery and not at moment of the diagnosis. As a consequence of that, our long-term results are remarkable better compared to other TC series. Secondly, the ESTS database included only cases coming from high-volume surgical centers and early-term results (i.e., high rate of radicality) may be not completely indicative of all surgical cases worldwide. There are no data available concerning the causes of death, including those related to MG exacerbation.

Finally, there is no pathological central revision of the surgical specimens and we cannot exclude the possibility that some tumors were composed from a mixture of thymoma B3 and a small area of thymic carcinoma. However, since only high-volume centers with large experience in thymic tumors pathology and surgery were involved in the present study, this risk is rather limited.

On the other hand, the present study has the great merit of focusing on a specific issue that was not well investigated so far. We have analyzed the data coming from a validated and largely recognized database (the ESTS database) which is the main strength of the present study. We also identified some clinico-pathological variables affecting long-term survival in our cohort of patients that could be considered in the daily clinical practice. Lastly, when investigating the association between MG and other clinico-pathological variables, we also observed two different phenotypes (elderly (>60 years) and male patients) that more frequently correlate with MG (as never reported before).

## 5. Conclusions

Despite less frequently than in thymoma patients, MG may occur in about 10% of TC-patients. In the ESTS database, young age, female gender, early Masaoka stage, and postoperative radiotherapy seem to positively affect long-term survival.

On the contrary, the coexistence of MG seems not to affect significantly survival in surgically treated TC-patients.

## Figures and Tables

**Figure 1 diagnostics-12-01764-f001:**
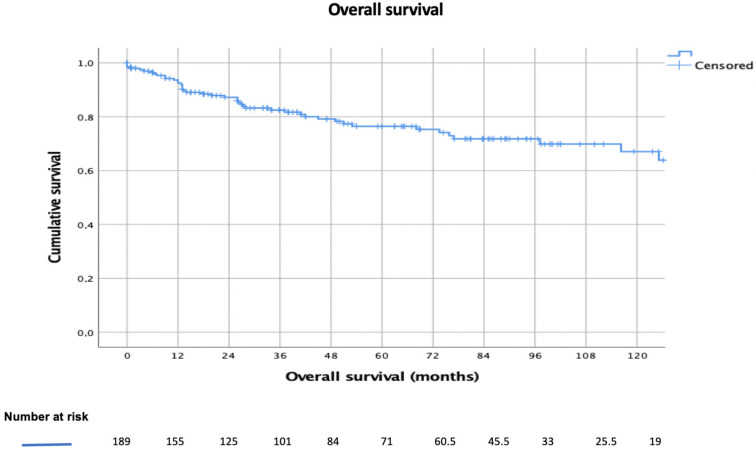
Overall survival of the entire population of thymic carcinomas extracted from the ESTS database.

**Figure 2 diagnostics-12-01764-f002:**
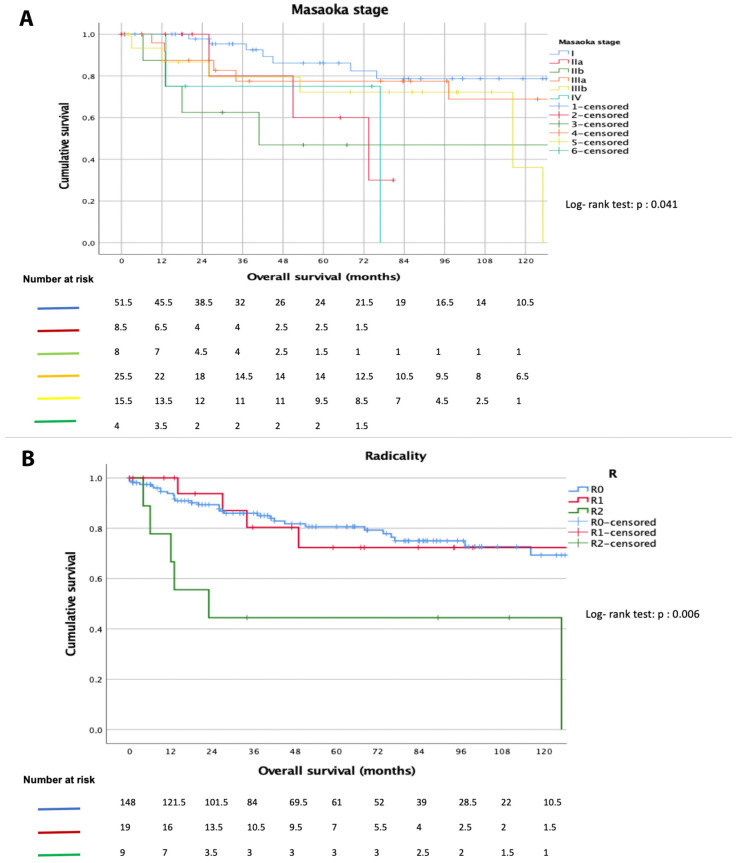
Survival curves according to Masaoka Stage (**A**) and radical resection status (**B**).

**Table 1 diagnostics-12-01764-t001:** Clinical and pathological characteristics.

Variables		MG Patients	Non-MG Patients	*p*-Value
		(#22)	(#181)	
Gender	Male	17 (77.3%)	79 (43.6%)	0.003
	Female	5 (22.7%)	102 (56.4%)	
Age ≥ 60 years	0	14 (63.6%)	75 (41.4%)	
	1	8 (36.4%)	106 (58.6%)	0.05
ECOG	0	2 (9%)	51 (28.2%)	0.29
	1	1 (4.5%)	34 (18.8%)	
	2	2 (9%)	9 (4.1%)	
	3	0 (0%)	3 (1.7%)	
	4	0 (0%)	2 (1.1%)	
	Missing data	17 (77.5%)	82 (46.1%)	
Neoadjuvant Therapy	0	18 (81.8%)	59 (32.6%)	0.61
	1	4 (18.2%)	18 (9.9%)	
	Missing data	0 (0%)	104 (57.5%)	
Surgical access	Sternotomy	8 (36.4%)	70 (38.7%)	0.33
	Thoracotomy	0 (0%)	33 (18.2%)	
	Clamshell	0 (0%)	2 (1.1%)	
	Emiclamshell	0 (0%)	5 (2.7%)	
	Sternotomy + Thoracotomy	0 (0%)	6 (3.3%)	
	VATS	2 (9.1%)	8 (4.4%)	
	RATS	2 (9.1%)	5 (2.8%)	
	Missing data	10 (45.4%)	52 (28.8%)	
Masaoka stage	I	7 (31.8%)	46 (25.4%)	0.03
	IIa	5 (22.7%)	26 (14.4%)	
	IIb	4 (18.2%)	32 (17.7%)	
	III	3 (13.6%)	54 (29.8%)	
	IVa	2 (9.1%)	15 (8.3%)	
	IVb	1 (4.6%)	8 (4.4%)	
Complete Resection	R0	18 (81.8%)	140 (77.3%)	0.85
	R+	4 (18.2%)	41 (22.6%)	
Adjuvant therapy	0	9 (40.9%)	89 (49.2%)	0.06
	1	13 (59.1%)	55 (30.4%)	
	Missing data	0 (0%)	37 (20.4%)	
Type of therapy	Chemotherapy	1 (4.5%)	4 (2.2%)	<<0.001
	Radiotherapy	7 (31.8%)	53 (29.3%)	
	Chemo-radiotherapy	1 (4.5%)	2 (1.1%)	

**Table 2 diagnostics-12-01764-t002:** Unadjusted and adjusted effect of clinical variables on overall survival. (HR: hazard ratio; CI: confidence interval). In bold significant variables. * Statistically significant.

Variables	Univariable Analysis		Multivariable Analysis	
	HR [95% CI]	*p*-Value	HR [95% CI]	*p*-Value
Female gender	0.6 [0.3–1.0]	0.063	0.4 [0.2–1.0]	**0.05 ***
Age < 60 years	**0.5 [0.3–0.9]**	**0.039 ***	0.1 [0.0–0.4]	**0.001 ***
Myasthenia Gravis	1.9 [0.9–4.5]	0.096	8.7 [0.8–95.3]	0.07
Minimally invasive approach	1.0 [0.2–4.6]	0.67		
Masaoka stage ≤ IIa	**0.2 [0.4–0.9]**	**0.04 ***	0.4 [0.2–0.8]	**<<0.001 ***
R0 resection	**0.7 [0.1–5.2]**	**0.006 ***	0.2 [0.0–2.1]	0.2
Adjuvant radiotherapy	**0.6 [0.3–1.1]**	**0.003 ***	0.5 [0.2–0.8]	**<<0.001 ***

**Table 3 diagnostics-12-01764-t003:** Unadjusted and adjusted effect of clinical variables on overall survival in patients undergone a radical thymectomy (R0). (HR: hazard ratio; CI: confidence interval). In bold significant variables.

Variables	Univariable Analysis		Multivariable Analysis	
	HR [95% CI]	*p*-Value	HR [95% CI]	*p*-Value
Female gender		0.197	0.3 [0.1–0.8]	0.017
Age < 60 years		**0.007 ***	0.1 [0.0-0.5]	**0.001 ***
Myasthenia Gravis		0.345		
Minimally invasive approach		0.834		
Masaoka stage ≤ IIa		**0.04 ***	0.7 [0.2–3.9]	**0.001 ***
Adjuvant radiotherapy		0.565		

* *p* < 0.017 (Bonferroni adjusted *p*-value).

**Table 4 diagnostics-12-01764-t004:** Surgical results in TC patients: an overview of pertinent literature.

	#	Radicality	5-Years OS	Variables Affecting Survival						
				Gender	Age	M.G.	Masaoka Stage	R0	Post-op RT	Post-op CHT
Weksler (2013)	290	56%	0%	X	-	-	X	X	-	-
Ruffini (2014)	229	69%	61%	-	-	-	X	X	X	-
Ahmad (2015)	1042	61%	60%	-	-	-	X	X	X	X
Li (2016)	49	61%	≈50%	-	-	X	-	X	-	-
Fu (2016)	329	58%	67%	-	-	-	X	X	X	-
Present series (2021)	203	78%	75%	X	X	-	X	X	X	-

## Data Availability

Data are available among the ESTS thymic registry.

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
