# Peer review of "Does Myasthenia Gravis Affect Long-Term Survival in Thymic Carcinomas? An ESTS Database Analysis"

_diagnostics, 2022, doi:10.3390/diagnostics12071764_

Round 1

Reviewer 1 Report

This study explored the effect of co-existing MG on the long term survival of patients with thymic carcinoma. Younger age, lower Masaoka stage and the use of adjuvant therapy (postoperative radiotherapy) were found to be favorable prognostic factors while the presence of MG did not appear to have an effect on the prognosis. 

This is a very good study. As the authors state, the study has a lot of positive aspects: The patients are from a well-established database (ESTS) with detailed information, the number of patients with thymic Ca and MG is relatively large (22 patients), multivariate analysis takes into consideration a lot of related factors.  

In the Abstract, results of univariate analyses are given. I think it is more important to give the results of mutivariate analysis. Also, could it be sufficient to just say they are significant instead of giving numbers? 

I could not understand the last sentence of the Discussion. Could they be clearer and explain what they mean? 

There are a few typos and awkward sentences:

MG is sometimes written as M.G.

Is fatigable weakness better than fatigability and weakness? 

‘Different from other registries…’ instead of ‘Differently by other registries…’

‘…the present population of study may not be representative of …’ instead of ‘…the present population of study may be not representative of …’ 

Possible awkward constructions: Despite no exact data…; the use of undergone (?who had undergone)

Author Response

Thank you for your kind revision.

R: In the Abstract, results of univariate analyses are given. I think it is more important to give the results of mutivariate analysis. Also, could it be sufficient to just say they are significant instead of giving numbers? 

A: Thank you for the advice. We agree with you and we modified the abstract accordingly.

R:I could not understand the last sentence of the Discussion. Could they be clearer and explain what they mean? 

A: Thank you for the advice. We explained better the concept in the last phrase.

R:

There are a few typos and awkward sentences:

MG is sometimes written as M.G.

Is fatigable weakness better than fatigability and weakness? 

‘Different from other registries…’ instead of ‘Differently by other registries…’

‘…the present population of study may not be representative of …’ instead of ‘…the present population of study may be not representative of …’ 

Possible awkward constructions: Despite no exact data…; the use of undergone (?who had undergone)

A: thank you. A further linguistic revision was done.

Reviewer 2 Report

The manuscript by F. Lococo et al. presents a detailed analysis of the possible link between Myasthenia gravis and thymic carcinoma. In my opinion, the study is well performed and presented.

I have some technical comments regarding the presentation of the graphs in this manuscript. Why the y axis is marked as (%) in Figure 1 and Figure 2 if the total scale is between 0 and 1? Graphs in Figure 2 could be oriented one below the other instead of side-by-side - they are hardly visible, especially in the printed version. If possible, please increase the resolution.

Author Response

Thank you for your nice revision.

Thank you for your comments. There was a typo in Figure 1 and 2 that we corrected. Furthermore, we accepted your advice to orient the figure one below the other and we improved the resolution up to 400 DPI.